# Impaired Cardiomyocyte Maturation Leading to DCM: A Case Report and Literature Review

**DOI:** 10.3390/medicina59061158

**Published:** 2023-06-16

**Authors:** Letao Zhou, Jinglan Huang, Hong Li, Hongyu Duan, Yimin Hua, Yuxuan Guo, Kaiyu Zhou, Yifei Li

**Affiliations:** 1Key Laboratory of Birth Defects and Related Diseases of Women and Children of MOE, Department of Pediatrics, West China Second University Hospital, Sichuan University, Chengdu 610041, China; 2Department of Nursing, West China Second University Hospital, Sichuan University, Chengdu 610041, China; 3Key Laboratory of Molecular Cardiovascular Science of Ministry of Education, Beijing Key Laboratory of Cardiovascular Receptors Research, Health Science Center, School of Basic Medical Sciences, The Institute of Cardiovascular Sciences, Peking University, Beijing 100191, China

**Keywords:** cardiomyocyte maturation, dilated cardiomyopathy, myocardium non-compaction, *ACTN2*, *RYR2*

## Abstract

*Background*: The maturation of cardiomyocytes is a rapidly evolving area of research within the field of cardiovascular medicine. Understanding the molecular mechanisms underlying cardiomyocyte maturation is essential to advancing our knowledge of the underlying causes of cardiovascular disease. Impaired maturation can lead to the development of cardiomyopathy, particularly dilated cardiomyopathy (DCM). Recent studies have confirmed the involvement of the *ACTN2* and *RYR2* genes in the maturation process, facilitating the functional maturation of the sarcomere and calcium handling. Defective sarcomere and electrophysiological maturation have been linked to severe forms of cardiomyopathy. This report presents a rare case of DCM with myocardial non-compaction, probably resulting from allelic collapse of both the *ACTN2* and *RYR2* genes. *Case Presentation*: The proband in this case was a four-year-old male child who presented with a recurrent and aggressive reduction in activity tolerance, decreased ingestion volume, and profuse sweating. Electrocardiography revealed significant ST-T segment depression (II, III, aVF V3-V6 ST segment depression >0.05 mV with inverted T-waves). Echocardiography showed an enlarged left ventricle and marked myocardial non-compaction. Cardiac magnetic resonance imaging revealed increased left ventricular trabeculae, an enlarged left ventricle, and a reduced ejection fraction. Whole exome sequencing revealed a restricted genomic depletion in the 1q43 region (chr1:236,686,454-237,833,988/Hg38), encompassing the coding genes *ACTN2*, *MTR*, and *RYR2*. The identified variant resulted in heterozygous variations in these three genes, with the *ACTN2* g.236,686,454-236,764,631_del and *RYR2* g.237,402,134-237,833,988_del variants being the dominant contributors to the induction of cardiomyopathy. The patient was finally diagnosed with DCM and left ventricular myocardial non-compaction. *Conclusions*: This study reports a rare case of DCM with myocardial non-compaction caused by the allelic collapse of the *ACTN2* and *RYR2* genes. This case provides the first human validation of the critical role of cardiomyocyte maturation in maintaining cardiac function and stability and confirms the key findings of previous experimental research conducted by our group. This report emphasizes the connection between genes involved in regulating the maturation of cardiomyocytes and the development of cardiomyopathy.

## 1. Introduction

The progression of heart development is a complex process [1]. Disorders of cardiac development can result in abnormal heart structure, congenital heart disease, and dysfunction in myocardial function, collectively referred to as cardiomyopathy. During fetal development, cardiomyocytes are primarily in a stage of differentiation and proliferation, with a limited duration of postnatal maturation. During this maturation process, cardiomyocytes undergo morphological, metabolic, and electrobiological adaptations to the internal and external environment, contributing to the formation of a mature and functional heart [1]. The cardiomyocytes then enter the adult phase, where they maintain contractile function throughout the lifespan. However, while protocols have been established for cardiomyocyte lineage differentiation and proliferation using progenitor cells or induced pluripotent stem cells (iPSCs) in vitro, the generation of mature and functional cardiomyocytes for tissue engineering and heart regeneration remains a challenge. The maturation of cardiomyocytes is a cutting-edge field in cardiovascular research and understanding the molecular mechanisms of cardiomyocyte maturation are crucial for advancing the understanding of cardiovascular disease. Current evidence demonstrates that the impairment of cardiomyocyte maturation may induce dilated cardiomyopathy (DCM) and left ventricular non-compaction (LVNC). Additionally, DCM and LVNC have been identified as inherited cardiomyopathy, and several variants in a series of critical genes participate in the causes of such cardiomyopathy [2], including *TTN* [3], *DES* [4], *LMNA* [5] and *RBM20* [6]. Moreover, the genetic background between DCM and LVNC is overlapped among a set of genes, which are mainly encoded in the protein molecules in sarcomere and Z-disc formation. Our research group has been studying the regulation principles of cardiomyocyte maturation for several years [1,7,8]. We have identified that sarcomere, mitochondrial, and calcium handling maturation are relatively independent, and that the optimal serum response factor (SRF) signaling in response to mechanical stress guides the entire process of cardiomyocyte maturation. Using CRISPR-Cas9 and associated adenovirus technologies, we studied the molecular function of actinin alpha 2 (*ACTN2*) and ryanodine receptor 2 (*RYR2*) in regulating cardiomyocyte maturation [7,8]. Our results showed that *ACTN2* and *RYR2* were essential for maturation, particularly for maintaining sarcomere function and electrophysiology, respectively [7,8]. In mosaic mouse models, single depletion of either *ACTN2* or *RYR2* preserved normal heart contractile function while avoiding the influence of heart failure on cardiomyocytes. However, high doses of *ACTN2* or *RYR2* depletion resulted in DCM and LVNC. The simultaneous loss-of-function impact of *ACTN2* and *RYR2* in human patients is rare, making it difficult to validate the findings of our experimental analyses.

In this report, we present a novel case of restricted genomic depletion in the 1q43 region (chr1: 236,686,454-237,833,988/Hg38), which includes the coding genes *ACTN2*, *MTR*, and *RYR2*. This variant resulted in heterozygous variants in these three crucial genes. The variants *ACTN2* g.236,686,454-236,764,631_del and *RYR2* g.237,402,134-237,833,988_del were found to contribute to cardiomyopathy in a dominant manner, while *MTR* is usually associated with neurodevelopmental disorders in a recessive manner. This is a rare case of heterozygous loss-of-function variants in both *ACTN2* and *RYR2* genes, offering an opportunity to study the clinical manifestation and phenotype when sarcomere and calcium handling maturation are impaired in cardiomyocytes. This rare case also demonstrated DCM with left ventricular myocardial non-compaction which may be due to *ACTN2* and *RYR2* variants, validating previous experimental findings. However, it is a pity that there is no way to build a similar mouse model to test the variants.

## 2. Case Presentation

### 2.1. Medical History and Physical Examination

This study had been approved by the Ethics Committee of the West China Second Hospital of Sichuan University (approval number 2014-034). Informed consent was obtained from the patient’s parents before performing whole exon sequencing and for the inclusion of the patient’s clinical and imaging details in subsequent publications.

This patient was a 4-year-old boy who was admitted to our hospital due to a decrease in activity tolerance for 8 days, as well as decreased ingestion volume and heavy sweating. The child also reported shortness of breath, persistent cough, palpitations, and a pale complexion. However, his parents reported no symptoms of cyanosis, fever, seizures, edema, vomiting, diarrhea, oliguria, or abdominal pain for this patient. The patient did not have any history of chest pain or syncope. At birth, the proband was full-term with a gestational age of 38 weeks and a birth weight of 2900 g. Although there was some growth retardation observed during development, the child’s cognitive and motor abilities were normal. On examination, the patient appeared acutely ill with severe diaphoresis and fatigue. His heart rate was 115 beats per minute and had irregular premature beats. Blood pressure was elevated in the left upper limb, but normal in the other three extremities. The patient also showed signs of slight malnutrition but had a normal response to external stimuli.

On physical examination, the patient had no trauma injuries and had symmetrical respiratory movements in both lungs. Breath sounds were rough, and a diffuse small wheezing was heard in both lungs. The apex of the heartbeat was located in the left lower quadrant and the heart boundary was slightly enlarged. The heart rhythm was heterogeneous and had occasional premature beats. Heart sounds were dull, and no murmurs were heard. The abdomen was soft and medium in texture, and the spleen was not palpable. Muscle strength and tension in all four extremities were normal, and there were no pathological signs or meningeal irritation.

The patient’s past medical history was unremarkable and there was no recent history of viral infection. The patient’s parents reported no history of illnesses, especially in the cardiovascular system, and no family history of cardiac attacks or cardiovascular diseases. There was no history of hypertension or coronary artery disease in the family.

### 2.2. Laboratory and Imaging Examinations

The initial laboratory results showed normal results for routine blood cell tests, blood gas analysis, hepatic and renal function. Elevation of B-type natriuretic peptide (139.65 pg/mL, n.v. <60 pg/mL) was observed, while the serum level of cTnI was within the normal range. Negative results were obtained for rheumatic screening, autoimmune antibodies, and thyroid function. Screening for potentially infectious viruses, including SARS-CoV-2, coxsackie virus, adenovirus, influenza virus, rhinovirus, and respiratory syncytial virus, all resulted in negative results. Electrocardiography (ECG) showed significant ST-T segment depression (II, III, aVF V3-V6 ST segment depression >0.05 mV with inverted T-waves, Figure 1A) and atrial (Figure 1B) and ventricular (Figure 1C) premature contractions were recorded with a Holter device. Additionally, the Prolonged QT interval is also reported. However, when recalculating the QTc using Bazett’s formula, it did not reach the criteria for long QT syndrome. Echocardiography revealed a dilated left ventricle (LVIDd = 48 mm, Z-score = 3.81, Figure 2A) with significant myocardial non-compaction (Figure 2A′), including a pseudotendon cord in the left ventricle. Chest computed tomography (CT) confirmed these findings of a dilated left ventricle and myocardial non-compaction (Figure 2B). A coronary artery CTA was performed, with images demonstrating normal structure and routine of the three branches of coronary arteries (Figure 2C,D). Cardiac magnetic resonance (CMR) showed an increased amount of left ventricular trabeculae, a dilated left ventricle (LVIDd = 47 mm), and a reduced ejection fraction (37.8%). The mass of the left ventricular myocardium was within the normal range (LV mass = 46.64 g, Z-score = −0.31), with no evidence of myocardial edema or significant fibrosis identified on the CMR examination (Figure 2E,F).

After a thorough series of examinations, myocarditis, aneurysms and structure malformation were ruled out as potential causes. Non-invasive ventilation support was initiated, and antibiotics were administered to treat a possible pulmonary infection. Patients with cardiomyopathy usually have some myocardial metabolic disorders, which are mostly caused by abnormalities in mitochondrial metabolism; thus, we used creatine phosphate and levocarnitine to provide myocardial protection treatment. Furthermore, medications such as digoxin, captopril, and metoprolol were used to support heart function and improved ventricular remodeling. Following two weeks of intensive care and treatment, the patient’s cardiac contractile function improved, and he was discharged with a plan for close monitoring.

### 2.3. Molecular Results

Based on the laboratory and imaging tests, a specific form of cardiomyopathy was strongly suspected. Subsequently, whole exon sequencing (WES) was performed to identify any crucial genetic variants. The WES analysis revealed a large-scale depletion of bases at the 1q43 site in the proband. The genomic location of the variant was located at chr1: 236,686,454-237,833,988/Hg38, encompassing the *ACTN2*, *MTR*, and *RYR2* coding genes (Figure 3A). The *ACTN2* gene lost exons 2 to 21, while the *RYR2* gene lost exon 1, and the *MTR* gene was completely lost in the variant allele. The proband’s parents were not carriers of the variant and did not display any cardiac-related clinical manifestations. Sanger validation was performed to confirm the absence of the variant. Furthermore, no other cardiovascular-related variants were found between the proband and her parents. According to the MutationTaster analysis, the variant was deemed a disease-causing mutation with probability values of 1.00 for *ACTN2* g.236,686,454-236,764,631_del and *RYR2* g.237,402,134-237,833,988_del (Figure 3B,C). The large-scale depletion has not been reported in the 1000 g and ExAC databases. The molecular crystal structure of *ACTN2* was built using AlphaFold (AF-P35609-F1, Figure 3D), revealing that the depletion of exons 2-21 would result in a truncated protein lacking CH1, CH2, EF-hand1, and EF-hand2 domains. The molecular crystal structure of *RYR2* was built using SWISS-Model (7ua3 for heteromer and 7u9x.1.A for monomer, Figure 3E), revealing that the depletion of exon 1 would lead to the interruption of transcription with a missing start codon. Therefore, the expression level of *RYR2* will decrease, but still be partially expressed due to the presence of the allele. This is a large deletion which began from the exon 2 of *ACTN2* and ended at the exon 1 of *RYR2*. As such, there was a sequence depletion in *ACTN2* loci and the promoter of *RYR2* prior to exon 1, which would definitely cause the loss of expression on the impair allele. However, the RYR2 is a large protein, and exon 1 is a small part. Furthermore, the depletion of promoter would certainly stop the transcription of the remaining sequencing.

According to the OMIM and ClinGen databases, the *MTR* gene is inherited in a recessive manner. Most reported cases of MTR-associated disorders are due to homozygous or compound heterozygous loss-of-function variants, suggesting that haploinsufficiency of the *MTR* gene is not a mechanism of disease. On the other hand, *ACTN2* and *RYR2* have been associated with cardiovascular disease through dominant inheritance patterns. Hence, haploinsufficiency should be considered for these two genes if loss-of-function alleles are identified. In the present case, the heterozygous variants in *ACTN2* (g.236,686,454-236,764,631_del) and *RYR2* (g.237,402,134-237,833,988_del) could account for the clinical manifestation in the proband. We believe that this should be the genetic cause of the disease in this proband.

### 2.4. Final Diagnosis and Treatment

After further evaluation, the patient was diagnosed with DCM and LVNC. To manage the condition, captopril, metoprolol, and digoxin were administered daily. Echocardiography and ECG were conducted every three months, and yearly cardiac magnetic resonance imaging (CMR) was performed. The follow-up results showed a persistently dilated left ventricle with non-compaction myocardium, with a left ventricular ejection fraction (LVEF) of 41%. Ultimately, heart transplantation was deemed the best alternative for the patient. 

## 3. Discussion

*ACTN2* and *RYR2* are genes that play crucial roles in maintaining the structure and function of cardiomyocytes. *ACTN2* mutations have been identified in patients with DCM and LVNC. These mutations disrupt the interaction between *ACTN2* and the Z-disk protein LIM domain myoglobin, leading to sarcomere dysfunction and regulation problems. This results in a loss of mechanical support and contractile ability in cardiomyocytes, contributing to the decline in their number and maturation, which manifests as LVNC and DCM. LVNC is a rare and severe hereditary cardiomyopathy characterized by prominent trabecular cavities and deep intrusions into the left ventricular cavity due to a failure in the embryonic myocardial compaction process [9]. Affected patients may experience right or left heart failure or complete heart failure and remain at risk for acute cardiac insufficiency. The types of heart failure caused by various cardiomyopathy are mainly associated with hemodynamic changes. Additionally, every cardiomyopathy has the possibility of acute cardiac insufficiency, even sudden cardiac death. For the proband, right ventricular dysfunction could be a secondary result of the left ventricular failure. In addition, congestive heart failure was considered as the main phenotype in acute phase. *ACTN2* and *RYR2* are both critical genes in cardiomyocytes [7,8]. According to previous studies from the Pubmed and NCBI database of inherited cardiovascular diseases, the mutations in *ACTN2* and *RYR2* serve as a dominant effect. Moreover, based on our previous research, the homozygous deletion of *ACTN2* or *RYR2* would induce a fetal-lethal condition. Thus, such two genes are of great importance, and haploinsufficiency had been identified in associated patients and animal models [9,10,11]. According to previous research, *ACTN2* is considered a major contributor to LVNC. In addition, *ACTN2* was mentioned as the causative gene in patients with myopathy in previous studies. Therefore, even though the proband was absent from any abnormality in muscle strength and muscle, we could not easily exclude any impairments of skeletal muscle system of the proband in this case, and scheduled follow-up was required. *RYR2* is a protein expressed on the sarcoplasmic reticulum (SR) in cardiomyocytes and is critical for releasing calcium ions. The most commonly associated cardiovascular disease with *RYR2* mutations is catecholaminergic polymorphic ventricular tachycardia (CPVT). These mutations lead to abnormal calcium handling in the endoplasmic reticulum, abnormal hyperphosphorylation of PKA and CaMKII binding sites, and increased sensitivity to adrenaline, which increases diastolic calcium levels and drives calcium exchange through the sodium–calcium exchanger (NCX1). This increased calcium exchange affects the stability of the cellular membrane potential and can trigger arrhythmia-related manifestations, such as CPVT. *RYR2* mutations have also been identified in some patients with cardiomyopathy, though the mechanism behind this association is not well understood. Given its dominant role in cardiomyocytes, further investigation is needed to confirm the role of *RYR2* in related cardiomyopathies. In this case, the proband’s level of *RYR2* expression has reduced, but the rest part of expression still acts on the Ca^2+^ ion channel. Thus, we supposed that this also has a certain impact on cardiomyopathy. As mentioned above, we cannot currently construct a similar mouse model to validate this event. In animal model research, the single base variant of *RYR2* would induce CPVT or deadly arrhythmias. However, large depletion of *RYR2* leads to severe DCM and impaired myocardial function [12,13,14]. Moreover, the animal models of *ACTN2* variant were also related with DCM and LVNC [7,11,15]. Accordingly, such previous reports supported the clinical phenotype of this patient was tightly correlated with the dysfunction of *ACTN2* and *RYR2*.

Based on the principles of cardiac development, maturation plays a crucial role in realizing the full biological function of cardiomyocytes. In recent years, our group and William T. Pu have identified several molecular mechanisms involved in regulating cardiomyocyte maturation [1,7,8,16]. There are three separate processes of maturation that occur in the domains of mitochondria/metabolism, sarcomere/morphology, and calcium handling/electrophysiology. Importantly, it has been noted that the mitochondrial, sarcomere, and electrophysiological functions are not fully established in the fetal stage and are mainly developed postnatally. As a result, certain genes associated with cardiomyopathy may not affect fetal development. However, these genes affect the process of fetal heart maturation, resulting in the failure of the myocardial compaction process in later embryonic development, thus leading to severe cardiomyopathy. In addition, the failure is also caused by external factors after birth, which is currently thought to be related to increased cardiac load or some connective tissue dysplasia syndromes. Commonly, the fetal onset LNVC was considered to be mainly due to the limited cellular number or impaired proliferation of cardiomyocytes. However, the postnatal LNVC could have resulted from the disorder of cardiomyocyte maturation, leading to a smaller cell size and cellular junction dysfunction. *ACTN2* and *RYR2* are two key molecules responsible for functional maturation of the sarcomere and calcium handling, respectively. In a study conducted by Guo et al., the regulation mechanism of *ACTN2* and *RYR2* was first revealed, and their specific role in cardiomyocyte maturation was documented [7]. The findings from subsequent studies confirmed that the collapse of the *ACTN2* and *RYR2* genes leads to severe cardiomyopathy. Loss of *ACTN2* results in a decrease in MRTF and impaired SRF signaling, leading to a disorder of morphological maturation [7]. Conversely, loss of *RYR2* leads to ER stress and the translocation of ATF4 and ATF6 into the nucleus, which promotes the expression of *DDIT3* and triggers autophagy-mediated protein degradation and cardiomyocyte atrophy [8]. Our experimental results have established the role of *ACTN2* and *RYR2* in cardiomyocytes. It is noteworthy that we encountered a human patient with loss-of-function mutations in both *ACTN2* and *RYR2*, who exhibited severe and early onset DCM. This clinical case validates the findings from our mouse studies. In addition, LNVC was associated with variants in other genes, including *TTN*, *DES* and *RBM20*. Variants in *TTN* in LVNC had been reported with a highly heterogeneous prevalence, and it had been confirmed that there was a haploinsufficient disease mechanism in titin truncation mutation cardiomyocytes, and that mitochondrial function would be impaired [17,18]. *DES* variants were identified in patients with an unexplained etiology of cardiomyopathy, and most of the *DES* variants were recorded in LNVC. Moreover, an assessment of the mitochondrial function in four probands heterozygous for a disease-causing *DES* variant confirmed a decreased metabolic capacity of mitochondrial respiratory chain complexes in myocardial muscle [19,20]. Furthermore, several reports had demonstrated the association between variants in *RBM20* and LVNC beyond DCM [21]. However, it was found that the same variant of *RBM20* would lead to both DCM and LVNC. Thus, the mechanisms of *RBM20* in regulating microstructure of cardiomyocytes required further analyses [22].

Previously, numerous reports have documented patients with 1q43 depletion. However, these patients displayed a large-scale sequence depletion in the region spanning from 1q42 to 1q44, which encompassed many coding genes. Most of these cases presented with neurological disorders as the primary manifestation. Studies by Thomas Rosto et al. [10] and Seiko Ohno et al. [14] have shown that a rare LVNC-CPVT overlap syndrome may be caused by a deletion in exon three of the *RYR2* gene. Although the proband in this case carries a duplication of exon 1 in *RYR2*, it is yet to be determined if it can also cause abnormalities in myocardial development and structure, leading to LVNC or LVNC-CPVT overlap syndrome. This highlights the importance of further follow-up. The findings suggest that loss-of-function variants of *RYR2*, beyond single base mutations, can induce cardiomyopathy. This case provides a unique opportunity to investigate a tightly restricted scale depletion between *ACTN2* and *RYR2* genes, which is restricted to myocardial disorders, and further our understanding of the maturation process in cardiovascular diseases.

## 4. Conclusions

This study presents a unique case of DCM with myocardial non-compaction, caused by the simultaneous collapse of *ACTN2* and *RYR2* genes. Cardiomyocyte maturation, which is regulated by *ACTN2* and *RYR2*, is critical for maintaining cardiac function and stability. The patient, who was diagnosed with this rare condition at a young age, highlights the importance of cardiomyocyte maturation in the development of cardiomyopathies. This case report is the first of its kind to validate the findings of our previous experimental research, emphasizing the crucial role of *ACTN2* and *RYR2* in cardiomyocyte maturation and the onset of cardiomyopathies. The report also sheds light on the critical genes involved in the regulation of cardiomyocyte maturation and their relationship with cardiomyopathies.

## Figures and Tables

**Figure 1 medicina-59-01158-f001:**
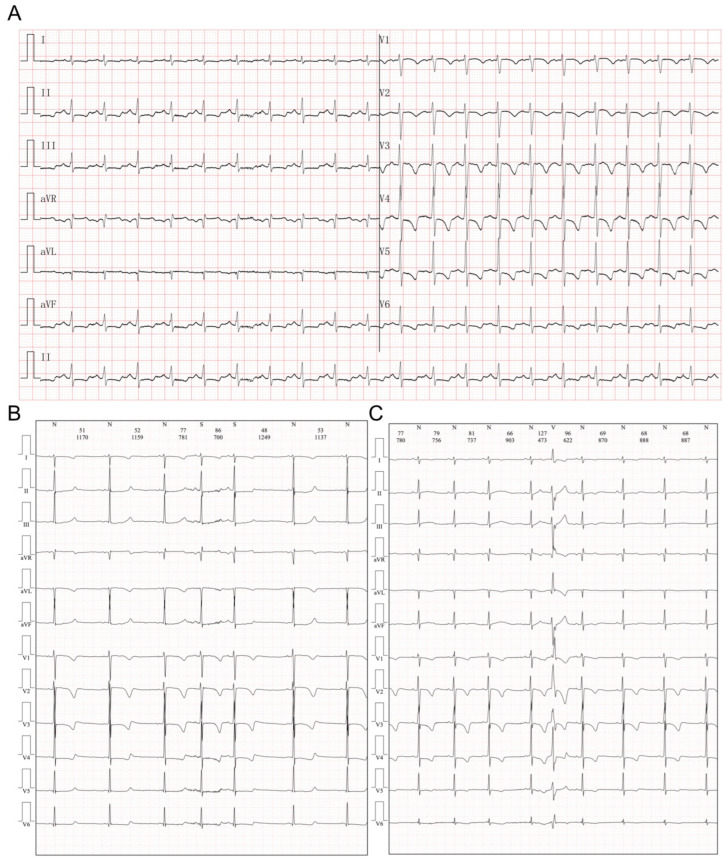
Electrocardiographic manifestation in the current proband. (**A**) Significant ST-T segment depression (II, III, aVF V3-V6 ST segment depression >0.05 mV with inverted T-waves). (**B**) Atrial premature contractions. (**C**) Ventricular premature contractions.

**Figure 2 medicina-59-01158-f002:**
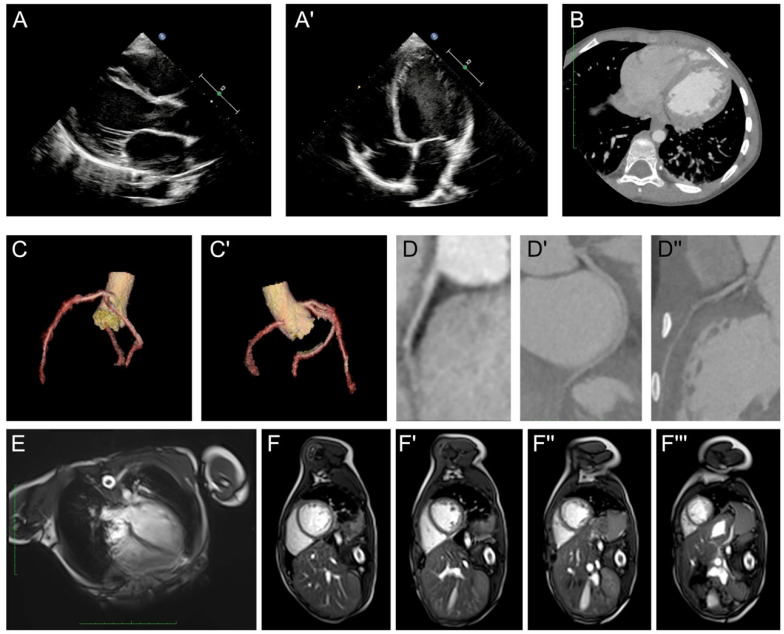
The proband’s clinical and radiographic manifestations. (**A**,**A′**) Echocardiography demonstrated an enlarged left ventricle (LVIDd = 48 mm, Z-score = 3.81) and myocardial non-compaction with a reduced ejection fraction. (**B**) Chest CT demonstrated the enlarged left ventricle and myocardial non-compaction. (**C**,**C′**,**D**,**D′**,**D″**) CTA demonstrated a standard structure and routine of three branches of coronary arteries. (**E**,**F**,**F′**,**F″**,**F‴**) CMR reported an increased amount of left ventricular trabeculae with enlarged left ventricle. No myocardial edema and significant fibrosis could be identified. LV indicated left ventricle.

**Figure 3 medicina-59-01158-f003:**
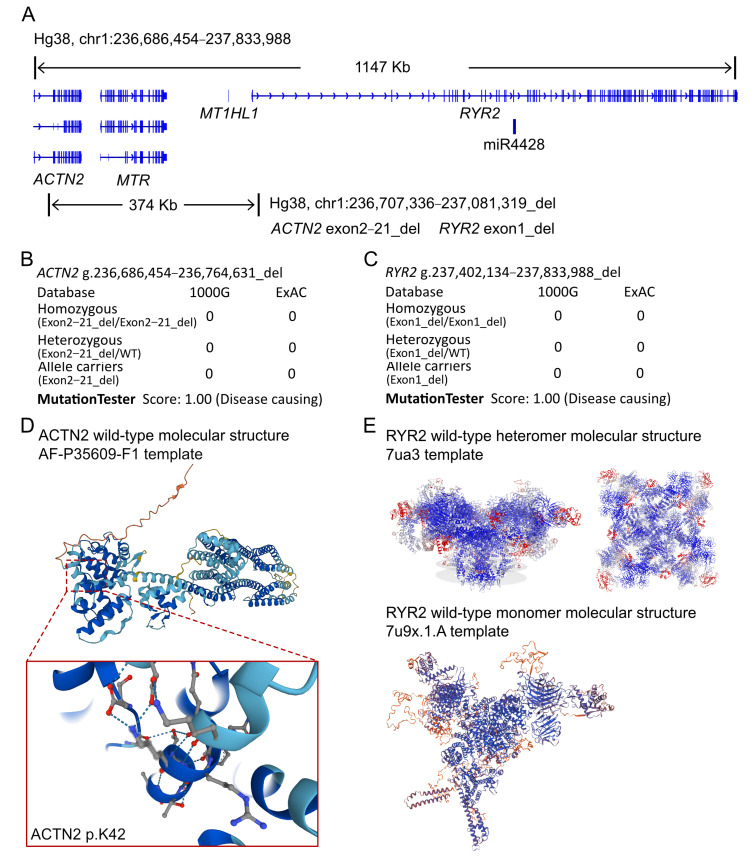
The genomic variant in this proband and molecular analysis. (**A**) The proband exhibited a heterozygous variant on chr1:236,686,454−237,833,988/Hg38. (**B**) Sanger sequencing validation. (**B**,**C**) The heterozygous variant of *ACTN2* g.236,686,454-236,764,631_del and *RYR2* g.237,402,134-237,833,988_del has never been reported in 1000 G and ExAC. Both of them were predicted to cause protein damaging by the MutationTaster analysis. (**D**,**E**) Protein structure predicted by AlphaFold and SwissModel for wild-type *ACTN2* and *RYR2*.

## Data Availability

Data sets used in this study are available from the corresponding author upon reasonable request.

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
