# Peer review of "Impaired Cardiomyocyte Maturation Leading to DCM: A Case Report and Literature Review"

_medicina, 2023, doi:10.3390/medicina59061158_

Round 1

Reviewer 1 Report

The manuscript “Collapse of ACTN2 and RYR2 Impaired Maturation of Cardio-myocytes Resulting in DCM: A Case Report and Literature Review” by Zhou L et al reports a rare cardiomyopathy case. It is interesting for the researchers in this field. However, the quality of the presentation is not good. The major problem is linked to the nature of the genomic deletion of patient. It is not clear, the data concerning the mutation such as the sequence of the junction of the deletion should be given in the figure 3 to assess the nature of mutation. It’s difficult to link the large deletion of RYR2 to the loss of only exon 1 and exon 2-21 for ACTN2. Is it one large DNA deletion or several deletions in the same chromosome? There is a doubt for the mutations from the information provided in the paper if no more complementary information is provided. The patient is heterozygous, it is strange why the WT allele is not functional, authors should explain or discuss this issue. Ideal is to provide dominant effect of these mutations. Figure 3 B, C D, E are less important than the data of the sequence mutation information and could be replaced. It is difficult to have the useful information from the figures 1 and 2 and their legends. No indications for ST-T segment depression, atrial and ventricular premature beats, no control electrocardiography to compare in the figure 1, legend is too simple.  Similar situation for figure 2, it is not informative. Taken together, the data presented is not so solid to support the conclusion.

Author Response

The manuscript “Collapse of ACTN2 and RYR2 Impaired Maturation of Cardio-myocytes Resulting in DCM: A Case Report and Literature Review” by Zhou L et al reports a rare cardiomyopathy case. It is interesting for the researchers in this field. However, the quality of the presentation is not good. The major problem is linked to the nature of the genomic deletion of patient. It is not clear, the data concerning the mutation such as the sequence of the junction of the deletion should be given in the figure 3 to assess the nature of mutation. It’s difficult to link the large deletion of RYR2 to the loss of only exon 1 and exon 2-21 for ACTN2. Is it one large DNA deletion or several deletions in the same chromosome? There is a doubt for the mutations from the information provided in the paper if no more complementary information is provided.

Response: Thanks for your kind suggestion. This is a large deletion which began from the exon 2 of ACTN2 and ended at the exon 1 of RYR2.  So that, there was a sequence depletion in ACTN2 loci and the promoter of RYR2 prior to exon 1, which would definitely cause the loss of expression on impair allele. Although the RYR2 is a large protein, and exon 1 is a small part. But the depletion of promoter would certainly stop the transcription of the remaining sequencing.   

The patient is heterozygous, it is strange why the WT allele is not functional, authors should explain or discuss this issue. Ideal is to provide dominant effect of these mutations.

Response: Thanks for your valuable comment. ACTN2 and RYR2 are both critical genes in cardiomyocytes. According to previous studies from Pubmed and NCBI database of inherit cardiovascular diseases, the mutations in ACTN2 and RYR2 serve as dominant effect. Besides, based on our previous research, the homozygous deletion of ACTN2 or RYR2 would induce fetal lethal. Thus, such two genes are of great importance, and haploinsufficiency had been identified in associated patients and animal models.

Figure 3 B, C D, E are less important than the data of the sequence mutation information and could be replaced.

Response: Thanks for your kind suggestion. We agree that such panels are not as important as panel A. But such panels help to provide to molecular information and structure of ACTN2 and RYR2, which indicated the impaired molecular function associated with reported mutations.

It is difficult to have the useful information from the figures 1 and 2 and their legends. No indications for ST-T segment depression, atrial and ventricular premature beats, no control electrocardiography to compare in the figure 1, legend is too simple.  Similar situation for figure 2, it is not informative.

Response: Thanks for your kind suggestion. However, we consider the figure 1 and figure 2 are quite important for a case report. In formatting a case report, the first step and the most essential part is to present the clinical phenotype, which would provide evidence to demonstrate the patient, we described, as a dilated cardiomyopathy. To perform an integrative assessment of a cardiomyopathy, the ECG, echocardiography and cardiac MRI are critical. So that, we provided solid information on the structure and function evaluation on heart, which could not be absent from a case report article. Besides, any ECG, echocardiography and cardiac MRI examinations did not require a comparison. As reference values and images had already been established based on large population and widely distribution across the world, it is not necessary to set up a control for a patient evaluation. In clinical practice, it would be easy to figure the differences in ECGs, echocardiography and cardiac MRI assessments. So that, we would be insistent to keep figure 1 and figure 2, as they are essential for a clinical case report study. Thanks for your kind comments to help us improve this manuscript.

Taken together, the data presented is not so solid to support the conclusion.

Response: Thanks for your kind help. And we have revised this manuscript according to your suggestion. We provided for explanation on the results, and we consider this would strength the results to support the conclusion.

Reviewer 2 Report

In the manuscript 'Collapse of ACTN2 and RYR2: Impaired Mutaration of Cardiomyocytes Resulting in DCM: A Case Report and Literature Review' submitted by Letao Zhou to Medicina, the authors screened a patient with DCM in combination with LVNC for genetic mutations and found a large mutation affecting the genes ACTN2, RYR2 and MTR. 

The manuscript is interesting and deserves from my point a view publication in Medicina. However, some points can improve the quality of this manuscript:

1.) Gene names should be written in the complete manuscript in Italics. 

2.) Sentence 1 of the Case Presentation: The is written in bold? In addition, male child should be exchanged against the word 'boy'.

3.) enlarged left ventricle = dilated left ventricle

4.) Could you please specifiy the ejection fraction? 

5.) I would shortly summarize within the introduction the genetic background of DCM and LVNC. In minimum, a good review article should be cited e.g. the following book chapter. Gerull, B., Klaassen, S., & Brodehl, A. (2019). The genetic landscape of cardiomyopathies. Genetic Causes of Cardiac Disease, 45-91. Please indicate the major genes for DCM (TTN, LMNA, RBM20, DES) and include relevant citations. I would also include the information that the genetic background of DCM and LVNC partially overlap.

6.) Discussion: I would include animal models in my discussion. Could you compare the described phenotype with your patient(s).

7.) Could you shortly discuss other LVNC genes like TTN, DES and RBM20 including relevant references for each gene?

8.) The figures are really nice and are fine.

In summary, I think the authors present an interesting case, where they discussed the specific phenotype leading to DCM / LVNC. I think the authors can fix the points mentioned before in a minor revision. Good luck with the revision. 

6.) 

Author Response

The manuscript is interesting and deserves from my point a view publication in Medicina. However, some points can improve the quality of this manuscript:

1.) Gene names should be written in the complete manuscript in Italics. 

Response: Thanks for your suggestion, and we have revised the manuscript according to your suggestion.

2.) Sentence 1 of the Case Presentation: The is written in bold? In addition, male child should be exchanged against the word 'boy'.

Response: Thanks for your suggestion, and we have revised this.

3.) enlarged left ventricle = dilated left ventricle

Response: Thanks for your suggestion, and we have revised this.

4.) Could you please specify the ejection fraction? 

Response: Thanks for your comment, and we have provided specific value of ejection fraction.

5.) I would shortly summarize within the introduction the genetic background of DCM and LVNC. In minimum, a good review article should be cited e.g. the following book chapter. Gerull, B., Klaassen, S., & Brodehl, A. (2019). The genetic landscape of cardiomyopathies. Genetic Causes of Cardiac Disease, 45-91. Please indicate the major genes for DCM (TTN, LMNA, RBM20, DES) and include relevant citations. I would also include the information that the genetic background of DCM and LVNC partially overlap.

Response: Thanks for your suggestion, and we have revised this.

6.) Discussion: I would include animal models in my discussion. Could you compare the described phenotype with your patient?

Response: Thanks for your suggestion, and we have revised this.

7.) Could you shortly discuss other LVNC genes like TTN, DES and RBM20 including relevant references for each gene?

Response: Thanks for your kind comment, and we have added the related information between the three genes you mentioned and LVNC in discussion part.

8.) The figures are really nice and are fine.

Response: Thanks for your comment.

Reviewer 3 Report

Congratulations to Authors for the important and excellent article!

Please check this minor comments and correct if needed:

Page2 Background

Line 7. :     Maybe I am not right, but I feel that the    term „In contrast” is not needed here.

                  It might be enough like this:          …..calcium handling. Defective sarcomere and electrophysiological…..

 Case presentation:

Line 10.:    236,764.631_del         please use the comma instead of point between 764 and 631

Abstract Conclusion (page2) :

Line 7:     please remove last point (doubled)         „development of cardiomyopathy.. „

Keywords:   Cardiomyocyte            letters remained in bold (not needed in bold)

Introduction:

Line 14:     Optional:  You might use plural at the words: „mechanism, disease” :

„molecular mechanism of cardiomyocyte maturation is crucial for advancing the understanding of cardiovascular disease”

„molecular mechanisms of cardiomyocyte maturation is crucial for advancing the understanding of cardiovascular diseases”

Line 27:       Using the „Since” (instead of „unfortunately”)  might be better:

„Unfortunately, the simultaneous loss-of-function impact of ACTN2”

„Since the simultaneous loss-of-function impact of ACTN2”

Line 33:     -236,764.631_del                point might be converted to comma:

                   -236,764,631_del

2.2. Laboratory and Imaging Examinations

Line 5:       „ potentially infected viruses”             infected might be converted to:

                      „potentially infectious viruses”  or simply  „potential viruses”

Line 27:                 „and they were discharged”         singular:       „and he was discharged”

2.3. Molecular Results

Line 12:  

-236,764.631_del                point might be converted to comma:

                   -236,764,631_del

Line 13:  

 Not (Figure 2B)

but:   (Figure 3B)

Line 15:  regarding SubFigure numbering, not 3C

AlphaFold (AF-P35609-F1, Figure 3C)

correct is:

AlphaFold (AF-P35609-F1, Figure 3D)

Line 18:          not 3D

7u9x.1.A for monomer, Figure 3D

correct is:

7u9x.1.A for monomer, Figure 3E

Line 27:

-236,764.631_del                point might be converted to comma:

                   -236,764,631_del

2.4     Final Diagnosis and Treatment

„…DCM with left ventricular myocardial non-compaction….”

please insert the acronym of LVNC after the word „non-compaction”    as some lines lower (Discussion Line 3)  the LVNC is used.

Figure 1. legend:

                  Figure 1. Electrocardiography manifestation               if agree, please correct to:

                  Figure 1. Electrocardiographic manifestation

Figure 2 legend:

               Figure 2. Clinical and radiographic manifestation of the current proband.

Some alternatives as suggestions only, not necessary to change to:

Figure 2.    The patient’s clinical and radiographic manifestations.

                   The proband’s clinical and radiographic manifestations.

                   The current patient’s clinical and radiographic manifestations.

                   The current proband’s clinical and radiographic manifestations.

                   The actual patient’s clinical and radiographic manifestations.

Other remark:   „And” is not needed:

„And no myocardial edema…. „            „ No myocardial edema……”

Figure 3 B

236,764.631_del         please use the comma instead of point between 764 and 631

Figure 3 legend:    The proband exhibited a heterozygou

                                 The proband exhibited a heterozygous

236,764.631_del         please use the comma instead of point between 764 and 631

had never reported in

had never been reported in

Please re-check this sentence:

Both of them were predicted protein damaging by

maybe:  Both of them were predicted to cause protein damaging by

TECHNICAL REMARK

Consent for publication

Please check:

(1 patient)

Not patients (1 patient):                               Not adults involved, please check:

„For patients who are under age of 16, written informed consent was obtained from the patient's parents for publication of this Case report and any accompanying images. And we also obtained the written informed consent from adults involved in this study for publication of related clinical data. A copy of the written consent is available for review by the Editor of this journal.”

Author Response

Dear reviewer,

I would express our great appreciate for your kind and detail review for this manuscript, which help us to correct a lot of mistakes in manuscript formation. All your suggestions were valuable for us to revise this manuscript. Thanks a lot. And we have revised the points you mentioned according to your comments.

As we involved the parents of this proband for sanger sequencing validation, we have obtained written informed consent from his parents (adults one). Thanks a lot.

Best,

Yifei

Round 2

Reviewer 1 Report

Dear Editor, I red the manuscript, but I do not find the real revision of manuscript. I hope that the case is real. If the mutation in ACTN2 is that (loss of exon2-21) described as the authors, the protein of ACTN2 will be only 40 amino acids if this mutated protein is stable, in this case the figure 3D is not correspond the situation. Similar for Figure 3E, if the promoter is removed, no protein RYR2 will be synthesized. So I am no satisfait with the response of authors.

Author Response

Dear Reviewer,
Thanks for your kind comment.
I am sorry that I might not express our idea clearly. 
As I mentioned in our manuscript, this patient carried a heterozygous variant which induced a large scale of genome depletion. We considered the mutation would lead to an expression collapse of ACTN2 and RYR2 in one allele, while another allele would be work well. So that, the ACTN2 and RYR2 contributed as a haploinsufficiency manner, and this case would not cause early death, some part of cardiomyocytes function would remain. 
Moreover, the figure 3D and 3E did not presented the mutant protein. We presented the wild-type protein structure in figure 3D and 3E to demonstrated the mutation caused which part of amino acids would be deleted. We did not show the truncated protein structure for each protein. The words in the figures were just to describe the results of such mutations. Also, in the figures, we clearly indicated the protein structures were wild-type.
Thanks for your kine help.
We hope we make ourselves clearly.
Best,
Yifei